# Molecular Profiling of Inflammatory Mediators in Human Respiratory Syncytial Virus and Human Bocavirus Infection

**DOI:** 10.3390/genes14051101

**Published:** 2023-05-17

**Authors:** Noorah A. Alkubaisi, Ibrahim M. Aziz, Asma N. Alsaleh, Abdulkarim F. Alhetheel, Fahad N. Almajhdi

**Affiliations:** 1Department of Botany and Microbiology, College of Science, King Saud University, Riyadh 11451, Saudi Arabia; 2Department of Pathology and Laboratory Medicine, College of Medicine, King Saud University, Riyadh 11451, Saudi Arabia

**Keywords:** pro-inflammatory cytokines, human orthopneumovirus, HRSV, human bocavirus

## Abstract

Infections due to human respiratory syncytial virus (HRSV) and human bocavirus (HBoV) can mediate the release of several pro-inflammatory cytokines such as IL-6, IL-8, and TNF-α, which are usually associated with disease severity in children. In this study, the change in the expression profile of cytokines and chemokines were determined during HRSV, HBoV, and HRSV coinfection with HBoV in 75 nasopharyngeal aspirates (NPAs) samples, positive real-time reverse transcriptase PCR Assay (rRT-PCR) for HRSV (*n* = 36), HBoV (*n* = 23) infection alone or HRSV coinfection with HBoV (*n* = 16). The samples were collected from hospitalized children. qPCR-based detection revealed that the levels of IL-6, IL-8, IL-10, IL-13, IL-33, and G-CSF were significantly (*p* < 0.05) greater in patients than in controls. IL-4, IL-17, GM-CSF, and CCL-5 were significantly elevated in children with HRSV coinfection with HBoV than in other groups (*p* < 0.05). TNF-α, IL-6, IL-8, IL-10, IL-13, and IL-33 in children with HRSV were significantly increased in severe infections compared to mild infections. Whereas, IL-10, IL-13, and IL-33 were significantly increased in severe infection in compared a mild infection in children with HBoV. Further large-scale investigations involving isolates are needed to enhance our knowledge of the association between viral infections and cytokine expression patterns during the different stages of HRSV and HBoV infection.

## 1. Introduction

The majority of lower respiratory tract infections (LRTI) in children worldwide are caused by the human respiratory syncytial virus (HRSV), formerly known as the human orthopneumovirus (HOPV) [1,2]. HRSV is a member of the genus *Orthopneumovirus* within the family *Pneumoviridae*, and order *Mononegavirales* [3]. HRSV is an enveloped virus that consists of a negative-sense single-stranded RNA genome with a 15.2-kb size. The genome contains 10 genes encoding 11 proteins that some of which have a significant role in both pathogenicity and classification [4]. The attachment protein (G), for instance, plays an important role in viral entry to the host. Based on the antigenic variations observed in it along with its interactions with monoclonal antibodies (mAbs), circulating viruses have been classified into two antigenic subgroups (A and B) [5,6]. As reported widely, both HRSV-A and B can co-circulate during an epidemic year with a slight predominance of type A [7]. Our group has previously shown that a similar pattern is occurring in Saudi Arabia. Both subtypes, HRSV-A and B, were circulating in some epidemic years with a slight predominance of HRSV-A [8,9,10]. Another etiological agent for LRTIs in infants and children is human bocavirus (HBoV). The virus, which was first identified in September 2005, was frequently detected in respiratory samples that were collected from children with acute wheezing episodes [11]. HBoV is a putative member of the family *Parvoviridae*, subfamily *Parvovirinae*, genus *Bocavirus*, that is spherical in shape, with icosahedral symmetry and non-enveloped nucleocapsid [11,12]. The virus is single-stranded DNA genome comprises three open reading frames (ORFs), the first two of which are nuclear phosphoprotein (NP1) and non-structural protein 1 (NS1), and the third of which is viral capsid proteins 1 and 2 (VP1 and VP2) [12]. There are four known human genotypes of this virus: type HBoV (1 to 4). HBoV1 is associated with LRTIs and, while the three additional genotypes HBoV2, 3, and 4 are associated with gastroenteritis and primarily seen in stools [13]. 

In response to localized inflammation infected by the virus, cytokines, such as interleukin-6 (IL-6), IL-8, tumor necrosis factor-α (TNF-α), CXCL2, and CXCL10 produced by various cell types in response to invading virus to promote the mobilization of macrophages and DCs can be linked to the severity of the illness in children [14,15,16]. It has been shown that children with HRSV ARTI have higher levels of pro-inflammatory cytokines including IL-4, IL-6, IL-10, and IL-13 [17,18,19,20,21]. According to another study, compared to influenza A virus infections, the concentrations of IL-4, IL-5, and CCL-5 are higher during HRSV with LRTI [22]. Furthermore, high levels of IL-6, IL-8, and IL-10 levels were correlated with disease severity in HRSV LRTI, the severity was determined by the requirement for supplemental oxygen [20,23,24]. The levels of IL-6, IL-1β, and IL-8 were also shown to be increased in patients infected with HRSV only or with both HRSV and human rhinovirus (HRV) [25]. Whereas, of HBoV-infected individuals TNF-a, IL-2, IL-5 IL-8, and IL-10 were found to be dramatically higher in HBoV-positive children in the acute stage of respiratory infection. [26,27]. 

Typically, cytokines have been evaluated by serological-based approaches including enzyme-linked immunosorbent assay (ELISA), Luminex, and cytometric bead array (CBA) [28]. However, cytokine mRNA quantification is widely used to investigate cytokine profiles via molecular-based assays, particularly in small samples. In comparison with serological-based approaches, real-time reverse transcriptase polymerase chain reaction (rRT-PCR) is the method of choice for rapid and reproducible, and flexibility to measure cytokine from cells, tissues, or tissue biopsies [29,30,31,32]. Therefore, the primary purpose of the current study is to identify the change in inflammatory mediators during HRSV, HBoV, and HRSV coinfection with HBoV. 

## 2. Materials and Methods

### 2.1. Study Population

A total of 75 nasopharyngeal aspirate samples (NPAs) were previously identified using rRT-PCR assay by King Khalid University Hospital (KKUH) (36 positives HRSV, 23 positives HBoV and 16 HRSV coinfection with HBoV). These samples were collected from infants and young children less than 5 years’ old. The patients were admitted to KKUH in Riyadh, Saudi Arabia during the winter seasons of 2019/20 and 2021/22 with acute respiratory symptoms including such as fever, dry cough, and fatigue. The number of days of supplementary oxygen was used to determine the severity for the purposes of this study. Patients in hospitals who required less than two days of oxygen supplementation were considered to have a severe illness. Patients hospitalized with 1 day of oxygen supplements or with no shortness of breath were classified as having a mild infection. Children who were reported positive for any other respiratory viruses were not included in the study. The healthy control NPAs samples (*n* = 25) were collected from infants without ALRTI symptoms and age-matched children of whom NPA samples were negative for HRSV and HBoV using rRT-PCR.

Control samples (*n* = 25) were collected from a group of healthy, asymptomatic age-matched children of whom NPA samples were negative for HRSV and HBoV using rRT-PCR. Clinical samples were collected following the protocols approved by the institutional review board (IRB) at King Khalid University Hospital, Riyadh (ref, No. 4/67/352665/IRB). 

### 2.2. Virus Studies

#### 2.2.1. Viral RNA Extraction

QIAamp Viral RNA Extraction Kit (Qiagen, Hilden, Germany) was used to extract viral RNA from clinical samples. HRSV and HboV were confirmed using one-step reverse transcriptase polymerase chain reaction (RT-PCR) Kit (Qiagen, Hilden, Germany) with the following set of primers forward primer HRSV-U-F (5′-GGAACAAGTTGTTGAGGTTTATGAATATGC-3′) and reverse primer HRSV-U-R (5′-CTTCTGCTGTCAAGTCTAGTACACTGTAGT-3′) [33]. 75 nasopharyngeal aspirate samples For the detection of HBoV, the same RT-PCR kit was used with the HboV-specific primer set: Boca-VP/NC-F (5′-AGCTGTGAGATTGTATGGGAAG-3′) and PanBoca-R (5′-AAAACAGCTCCCCCCACAAT-3′), the final 1/10 of the VP1 gene’s 3′ and the last 1/4 of the 3′ of the non-coding regions were targeted using this primer set, which flanks a 378-bp highly conserved sequence [34]. The reaction was performed in a GeneAmp 9700 thermal cycler (GeneAmp PCR system 9700, Applied Biosystems, Waltham, Massachusetts, United States) using the following cycling conditions: reverse transcription step at 50 °C for 30 min followed by initial denaturation at 95 °C for 15 min then 35 cycles of denaturation at 94 °C for 30 s, annealing at 52 °C for 30 s, and extension at 72 °C for 2 min; and a final extension step of 72 °C for 10 min. The amplified products were compared to a 100 bp DNA ladder (Qiagen, Hilden, Germany) and observed on an agarose gel stained with 1% ethidium-bromide.

#### 2.2.2. Quantitative Real-Time PCR

RNeasy^®^ Mini Kit (Qiagen, Hilden, Germany) was used to extract total RNA from clinical samples in accordance with the manufacturer’s instructions. The quantification of cytokines and chemokines in NPAs was determined by performing a quantitative qPCR using one step RT² SYBR^®^ Green/ROX™ qPCR Master Mix. The reactions were carried out in 7500 Fast real-time PCR (7500 Fast; Applied Biosystems, Waltham, MA, USA) using the following cycling conditions: reverse transcription at 37 °C for 15 min, initial PCR denaturation at 95 °C for 10 min, followed by annealing at 60 °C for 30 s, and extension at 72 °C for 30 s. The data were expressed as mean fold changes ± standard error for three independent amplifications. The sequences of the utilized primers used in PCR are presented in Appendix A [35,36,37,38,39,40,41,42,43,44,45,46,47,48,49].

#### 2.2.3. Analysis of Real-Time PCR Array Data

All arrays were run under the same conditions and Ct values were obtained using a constant baseline threshold for all PCR runs. All samples were analyzed in duplicate at least three separate experiments. Melting curve analysis was done to confirm the specificity of amplification and the lack of primer dimers. The results were obtained using the 2^−ΔΔCq^ method [50], and delta Cq (ΔCq) values obtained for the different genes were normalized based on the value of GAPDH amplified from the same genes and the fold-change in expression was calculated as referenced to the expression of the control (non-infected cells or negative control).

### 2.3. Statistical Analysis

The non-parametric one-way analysis of variance (ANOVA) with posthoc Dennet’s test was used for data analysis). Categorical data were presented as frequencies and percentages (%). Non-Gaussian variables were presented as the median. Independent Sample *t*-test and The Chi-Square independence test were used to indicate differences as all variables are nominal. The results with *p* < 0.05 were considered significant.

## 3. Results

### 3.1. General Patient Characteristics

Infants under one year of age were found to be the age group most affected, accounting for 44.5%, 65.2%, and 50.0% of all cases of HRSV, HBoV, and HRSV coinfection with HBoV, respectively. Based on gender, HRSV with 61.1% and HBoV with 60.9% were predominant in males than in females. Based on severity were found 55.6%, 78.3% of HRSV and HBoV (Table 1).

### 3.2. Cytokine and Chemokine Profile

IL-6, IL-8, IL-10, IL-13, IL-33, and G-CSF in the NPA were significantly greater (*p* < 0.05) in children with HRSV, HBoV and HRSV coinfection with HBoV than in controls (Table 2). 

(Table 3) showed that children with HRSV had significantly higher levels of IL-4, IL-13, IL-17, GM-CSF, and CCL-5 (*p* < 0.05) than children with HBoV alone or HRSV coinfection with HBoV, whereas, children with HBoV infection had significantly greater levels of TNF-α, IL-2, IL-6, IL-8, IL-10, IL-33, and CCL-3 compare to HRSV alone or HRSV coinfection with HBoV. IL-4, IL-17, GM-CSF, and CCL-5 were significantly elevated (*p* < 0.05) in children with HRSV coinfection with HBoV than HRSV or HBoV alone.

We next compared the proinflammatory cytokines and chemokines markers in the NPAs among children with HRSV and/or HBoV and disease severity. As depicted in (Figure 1), TNF-α, IL-6, IL-8, IL-10, IL-13, and IL-33 in children with HRSV were significantly increased in severe infection compared to mild infection (*p* < 0.05). Whereas, IL-10, IL-13, and IL-33 were significantly (*p* < 0.05) increased in severe infection compared to a mild infection in children with HBoV. TNF-α, IL-10, IL-13, and IL-33 were significantly (*p* < 0.05) increased in severe infection comparison to a mild infection in children with HRSV coinfection with HBoV.

## 4. Discussion

HRSV is the major cause of LRTI hospitalizations in young children, immunocompromised adults, and the elderly. Currently, there is no effective treatment or vaccine available [51]. Several pro-inflammatory cytokines can be released by innate immune cells during severe lung responses to RSV infection leading to lung immunopathology [52]. Concurrently, pro-inflammatory cytokines (e.g., CXCL8, CCL-3, CCL-2, and CCL5) and chemokines (e.g., IL-6, IL-8, IL-33, TNF-) are induced and secreted to the extracellular medium, which can be related to the severity of the disease in children. Therefore, the pro-inflammatory cytokines mediator may represent a strategy that contributes to HRSV pathogenesis and is associated with the disease severity of respiratory viral infections in children. In this report, the pro-inflammatory cytokines elicited by HRSV and HBoV were analyzed. In addition, to identify the correlation between these biomarkers and the development of lung fibrosis and the severity of immunopathological responses and comparison of pulmonary and in vitro gene expression.

Infants under one year of age were found to be the age group most affected, comprising 44.5% of all cases of HRSV. Males had a higher prevalence of HRSV than females according to gender. According to our earlier research, HRSV-A was somewhat more prevalent than HRSV-B between 2008 and 2016 in Saudi Arabia, causing lower respiratory tract infections in hospitalized children at a rate of 19.3 to 45.4% [8,9,10,53]. A similar predominance of HRSV-A over HRSV-B was reported earlier in Germany [54], Spain [55], France [56], Kuwait [57], and Lebanon [58]. Infants < 6 months of age were found to be the age group most affected age group, accounting for 59.32% of all cases. According to gender, men (18%) had more cases of HRSV than women (7%) [10].

The age group most afflicted, accounting for 59.32% of all cases, was infants under 6 months old. Males (18%) had a higher prevalence of HRSV than females (7%) when comparing the two genders.

In this study, qPCR-based detection revealed that the levels of IL-6, IL-8, IL-10, IL-13, IL-33, and G-CSF were significantly (*p* < 0.05) greater in children with HRSV, HBoV and HRSV coinfection with HBoV than in controls. TNF-α, IL-6, IL-8, IL-10, IL-13, and IL-33 in children with HRSV were significantly increased in severe infection compared to mild infection (*p* < 0.05) in children with HRSV. Children with HRSV have higher amounts of IL-8, IFN-a, IL-6, IL-3, IL-33, IL-1a, IL-2, IL-22, G-CSF, CCL-2, and CCL-3, which were associated with children’s disease severity [59,60,61]. According to Sheeran et al., all children with HRSV infection had significantly higher levels of IL-6, IL-8, and IL-10 in their nasal wash and tracheal aspirate samples than did control children [62]. Another study indicated that patient NPAs had higher levels of cytokines (IL-1, IL-2, IL-12, IFN-, TNF-a, IL-13, IL-4, IL-6, IL-10), chemokines (IP-10, IL-8, MIP1-a, MIP-1), growth factors (FGFb, PDGFbb, G-CSF), and cytokines (IL-1RA, IL-17) in nasal washes [18]. Further, early research also revealed elevated IL-4 levels, particularly in very young newborns, during RSV infection. However, another study found that only children with acute RSV-induced bronchiolitis had significantly elevated levels of IL-l0, but not IL-12 or IFN, possibly explaining the ineffectiveness of the cell-mediated immune response. According to other studies, nasal washes include high levels of IL-4, IL-6, IL-9, IL-10, and IL-13 [63,64,65], and in the lung in children with HRSV LRTI [66].

The current study also reported that IL-10, IL-13, and IL-33 were significantly increased in severe infection compared to mild infection in children with HBoV. Early study reported that cytokines including GM-CSF, Leptin, IL-1β, TNF-α, IL-8, IL-16, CCL-5, and IL-3 were clearly upregulated in the HBoV-positive cohort [67]. This study also revealed that IL-10 and IL-33 were markedly elevated in children with HBoV as well as with HRSV coinfection with HBoV coinfection. Further, IL-4, IL-17, GM-CSF, and CCL-5 were significantly elevated in children with HRSV coinfection with HBoV than in other groups (*p* < 0.05). Early research revealed that two or more respiratory viruses are co-infected in 10–30% of individuals under age 5 years [68,69,70]. According to a previous investigation, patients with HRSV coinfection with HBoV had greater rates of pneumonia than those with HBoV or HRSV infection alone [71].

IL-6 is a soluble mediator that is produced by macrophages and epithelial cells that has a pleiotropic influence on inflammation and immune response [72,73]. IL-6 is frequently associated with innate and adaptive immunity and regulates antimicrobial defense by promoting the differentiation of naive CD4^+^ and CD8^+^ T cells. During HRSV infections, Tabarani et al. reported that the level of IL-6 was evaluated in NPAs samples from children with HRSV. It’s interesting to note that among other inflammatory mediators (IFN-α, CCL3, CCL-4, and CCL-2), they have discovered an association between the severity of the clinical manifestations elicited by HRSV infection and high levels of IL-6 [74]. However, Brown et al. have proposed that high plasma levels of IL-6 could suggest a higher likelihood of infant hospitalization and severe bronchiolitis caused by HRSV [75]. Other studies demonstrated a positive correlation between high levels of IL-6 found in the NPAs of HRSV-infected patients and more severe forms of the disease, and that at the period of hospital admission, the levels of IL-6 in the serum could be used as indicators of the severity in patients with HRSV-caused LRTI [24,59].

The chemotactic factor known as IL-8 is known to be released by activated monocytes and macrophages and to encourage the directional migration of neutrophils, basophils, and T lymphocytes [76]. Disease severity caused by HRSV infection has been closely correlated with elevated IL-8 levels in NPAs samples, including the risk of mechanical ventilation [75,77]. Significantly, children’s higher NPA IL-6 and IL-8 levels have been associated with more severe HRSV infections [78]. Another study reported that the elevated concentrations of IL-8 in NPAs washes were associated with severe HRSV infection [74]. Similarly, the levels of IL-8 and CCL-5 were elevated in plasma and NPAs with disease severity in HRSV-infected children as compared to children with mild or moderate disease [79]. 

IL-33 is a member of the IL-1 family, which is constitutively produced from the structural and lining cells including fibroblasts, endothelial cells, and epithelial cells of the skin, gastrointestinal tract, and lungs that are exposed to the environment [80]. In respiratory tract epithelia, the main function of this cytokine is involved in early immune and inflammatory responses following HRSV infection. IL-33 is constitutively expressed by endothelial and epithelial cells. Unlike other interleukin 1 family members, IL-33 has been associated with the initiation and development of the innate and adaptive Th2-type immune response [81]. Clinically, patients with mild to moderately severe asthma show increased nasal IL-33 levels following HRSV inoculation and this increase correlates with symptom severity and viral load [82]. 

It is believed that IL-5 and IL-13 play important roles in the etiology of asthma, with IL-13 acting on a different pathway from IL-5 to cause airway hyperreactivity (AHR) in the allergic lung [83]. Importantly, IL-13 is generated and secreted by a variety of cells, including alveolar macrophages, basophils, mast cells, eosinophils, and CD4+ T cells, in response to IL-33 signaling. IL-33 directly stimulates in response to IL-33 signaling produced and secreted through various cells, including alveolar macrophages, and basophils [81]. Infants hospitalized with HRSV-caused bronchiolitis had high levels of IL-13 and IL-33, which were associated with the need for ventilation [65].

We acknowledge the shortcomings and limitations in terms of the methodology we used in our present study. Since it is easier to measure mRNA levels across the entire genome than protein levels, mRNA expression levels are frequently used as a proxy for estimating functional differences that occur at the protein levels [84]. Therefore, further work should be measured on HRSV and/or HBoV at protein levels of proinflammatory cytokines and chemokines markers for disease severity. In addition, rRT-PCR, which is becoming the method used for respiratory diagnosis, yields result as cycle threshold (CT) values, which is a semi-quantitative value that can broadly categorize the concentration of viral genetic material in a patient sample, therefore, further research is required to examine the relationship between viral load or CT values and inflammatory markers. Since in vivo study is not sufficient for proving the induction of chemokines and cytokines by HRSV and/or HBoV, in vitro model is required to support the in vivo observations.

## 5. Conclusions

In conclusion, this is an excellent and very promising study and really well done to identify the change in inflammatory mediators during HRSV, HBoV, and HRSV coinfection with HBoV. Results from this study revealed that qPCR-based detection revealed that the levels of IL-6, IL-8, IL-10, IL-13, IL-33, and G-CSF were significantly (*p* < 0.05) greater in children with HRSV, HBoV, and HRSV coinfection with HBoV than in controls. TNF-α, IL-6, IL-8, IL-10, IL-13, and IL-33 in children with HRSV were significantly increased in severe infections compared to mild infections. Whereas, IL-10, IL-13, and IL-33 were significantly increased in severe infection compared to mild infection in children with HBoV. IL-4, IL-17, GM-CSF, and CCL-5 were significantly elevated in children with HRSV coinfection with HBoV than in other groups (*p* < 0.05). Further investigations including patients during the various stages of HRSV infection are needed to understanding our knowledge of the biological impact of pro-inflammatory cytokine and cytokine expression patterns. 

## Figures and Tables

**Figure 1 genes-14-01101-f001:**
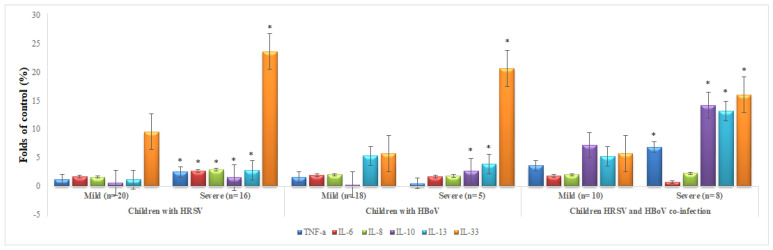
Comparison of cytokines and chemokines markers in the NPAs among children with HRSV and/or HBoV and disease severity. Note: Data presented Median (25th–75th) percentile. * Denotes a significant difference between mild and HBoV of each virus respectively at 0.05 level.

**Table 1 genes-14-01101-t001:** General patient characteristics.

Parameters	Children with HRSV *n* (%)	Children with HBoV *n* (%)	Children with HRSV Coinfection with HBoV *n* (%)	Control *n* (%)
Numbers	36	23	16	25
Age in years				
<1	16 (44.5)	15 (65.2)	8 (50.0)	11 (44.0)
1–3	11 (30.5)	5 (21.8)	5 (31.2)	9 (36.0)
3–5	9 (25.0)	3 (13.0)	3 (18.8)	5 (20.0)
Gender				
Male	22 (61.1)	14 (60.9)	12 (75)	14 (56.0)
Female	14 (38.9)	31 (39.1)	4 (25)	11 (44.0)
Type of virus				
HRSV-A	24 (66.7)			
HRSV-B	12 (33.3)			
Severity				
Mild	20 (55.6)	18 (78.3)	10 (62.5)	
Severe	16 (44.4)	5 (21.7)	6 (37.5)	

Note: Data presented as number (%).

**Table 2 genes-14-01101-t002:** Comparison of cytokine and chemokine profile between 75 patients and 25 healthy controls.

	Healthy Controls *(n* = 25*)*	Patients *(n* = 75*)*	*p* Value
INF-G	2.36 (1.8–4.18)	2.32 (0.81–4.17)	0.922
TNF-a	1.51 (0.38–0.80)	1.03 (0.74–6.10)	0.222
IL-1a	2.77 (1.44–34.9)	2.96 (0.92- 44.9)	0.452
IL-2	19.5 (0.53–27.0)	18.5 (0.53–41.0)	0.626
IL-4	1.39 (0.33–0.54)	1.56 (0.13–0.67)	0.455
IL-6	1.50 (0.49–1.43)	1.26 (0.49–4.32)	0.015
IL-8	0.53 (0.27–1.50)	1.42 (0.27–3.02)	0.011
IL-10	0.11 (0.26–1.87)	0.20 (0.13–2.87)	0.007
IL-13	0.70 (0.50–1.56)	3.28 (0.50–13.3)	0.018
IL-17	0.80 (0.80–1.56)	0.16 (0.05–1.56)	0.336
IL-22	2.23 (1.38–3.56)	2.55 (0.16–7.96)	0.525
IL-33	5.68 (7.1–12.01)	5.89 (3.73–37.01)	0.021
G-CSF	2.9 (2.64–60.03)	3.15 (1.55–61.0)	0.030
CCL-2	8.30 (0.35–13.7)	10.77 (0.35–22.45)	0.851
CCL-3	2.33 (0.35–0.12)	2.38 (0.78–0.35)	0.551
GM-CSF	1.10 (0.68–1.93)	1.18 (0.68–2.72)	0.737
CCL-4	1.89 (0.41–2.22)	2.10 (0.23–11.2)	0.801
CCL-5	0.18 (0.23–0.53)	0.11 (0.07–1.09)	0.22

Note: Data presented Median (25th–75th) percentile. *p*-value significant at 0.05 and 0.01 level.

**Table 3 genes-14-01101-t003:** Comparison of cytokine and chemokine profile between HRSV and HBoV infected patients.

Parameters	HRSV (*n* = 36)	HBoV (*n* = 23)	HRSV Coinfection with HBoV (*n* = 16)	*p*-Value
INF-G	2.32 (0.81–4.17)	2.12 (1.16–3.18)	3.70 (2.91–8.34)	0.450
TNF-a	1.94 (0.9–4.93)	1.51 (0.38–0.80)	1.87 (1.50–5.83) ^B^	0.015
IL-1a	3.11 (1.45–10.16)	3.91 (2.44–34.86)	24.47 (4.88–44.90)	0.161
IL-2	18.50 (3.39–41.0)	21.0 (0.53–27.0)	45.55 (24.06–64.25) ^B^	0.012
IL-4	1.56 (0.13–0.67)	1.40 (0.33–0.54)	1.90 (0.67–1.0) ^AB^	0.001
IL-6	1.26 (0.50–4.32)	1.67 (0.49–1.43)	2.31 (1.35–4.50) ^B^	0.008
IL-8	1.42 (0.70–3.02)	1.80 (0.27–1.16)	2.65 (1.45–4.75) ^B^	0.024
IL-10	1.20 (0.13–0.67)	1.63 (0.26–2.87) ^A^	1.06 (0.51–1.51) ^A^	<0.001
IL-13	3.28 (0.82–13.34)	1.90 (0.50–1.56)	4.05 (2.93–14.25) ^B^	0.017
IL-17	1.16 (0.08–0.75)	1.13 (0.05–0.13)	1.16 (1.08–1.95) ^AB^	<0.001
IL-22	2.55 (0.16–7.96)	2.15 (1.38–3.56)	4.55 (2.26–11.46)	0.196
IL-33	5.89 (3.73–12.34)	25.58 (7.07–37.01) ^A^	30.75 (17.05–37.82) ^A^	<0.001
G-CSF	3.15 (1.55–61.0)	28.9 (2.64–63.03)	28.53 (9.31–87.0)	0.139
CCL-2	10.77 (1.96–22.45)	6.69 (1.38–13.69)	18.19 (8.32–37.38)	0.198
CCL-3	2.38 (0.78–5.17)	4.33 (0.35–24.02)	14.96 (5.10–28.78) ^B^	0.002
GM-CSF	1.18 (0.73–2.72)	0.88 (0.68–1.93)	3.81 (2.42–4.78) ^AB^	<0.001
CCL-4	1.37 (0.18–8.46)	0.89 (0.41–2.22)	8.28 (3.0–12.37) ^B^	0.119
CCL-5	1.11 (0.07–1.09)	1.28 (0.23–0.53)	2.71 (1.11–2.25) ^AB^	0.001

Note: Data presented Median (25th–75th) percentile. *p*-value significant at 0.05 and 0.01 level. Superscript ^A & B^ presented posthoc analysis between HRSV and HBoV respectively.

## Data Availability

All the data is provided.

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
