# Peer review of "Molecular Profiling of Inflammatory Mediators in Human Respiratory Syncytial Virus and Human Bocavirus Infection"

_genes, 2023, doi:10.3390/genes14051101_

Round 1

Reviewer 1 Report

Background: The authors studied children under 5 years old, confirmed positive for HRSV and/or HBoV, admitted with acute respiratory symptoms to King Khalid University Hospital in Riyadh, Saudi Arabia during the winter seasons of 2019/20 and 2021/22. Control NPAs were obtained from 25 healthy, asymptomatic age-matched children. The primary aim of the study is to use qPCR to identify potential biomarkers of HRSV and HBoV infection disease severity.

Major Comments:

1.      The authors do not provide information about the length of time between symptom onset, hospital admission and first sample collection and the interval between infection and sampling is unclear. Responses are presumably time-based- was there any evidence for this?

2.      The secondary aim was to identify “whether the association of HRSV and HBoV would worsen the progression of the illness during HRSV and HBoV infections”, which was not addressed directly. The authors measured the levels of inflammatory mediators in patients with co-infection, but this was not associated with disease severity nor the progression of illness, particularly as the analysis is static and there is no mention of repeat sampling.

3.      The tables include individual cases in several categories, presenting the same data repeatedly in different ways. This could be described as ‘reshuffling the deck’, looking for patterns each time the deck is redistributed. Statistical advice should be sought.

Additional points:

1.      How were these controls recruited and consented?

2.      I could not see if all the datasets were from NPA or if some were serum/plasma. Please make this clear for each data table/graphic.

3.      Validation by direct measurement of selected protein level would provide confirmation.

4.      The methods say that the subtyping of RSV-A and -B was done by measuring the RSV G gene by RT-qPCR, but the paper cited examined the RSV N gene.

5.      The numbers in the tables do not always add up as they should.

I recommend:

1.      Table 1 should be online.

2.      Table 2 can remain in the main text.

3.      Table 3-7 should be online, replaced in the main text by visualisations of distribution of significant mediator values. Showing these data as, for example, Violin plots would show individual datapoints and the spread of values between groups.

It would be normal to allow the reader to assess the paper based on its strengths rather and saying, “this is an excellent and very promising study and really well done”.

Author Response

Response to reviewers-1’ comments

We are very grateful for the reviews and valuable comments provided by the editors and each of the external reviewers of this MS

Reviewer 1

Comments and Suggestions for Authors

Background: The authors studied children under 5 years old, confirmed positive for HRSV and/or HBoV, admitted with acute respiratory symptoms to King Khalid University Hospital in Riyadh, Saudi Arabia during the winter seasons of 2019/20 and 2021/22. Control NPAs were obtained from 25 healthy, asymptomatic age-matched children. The primary aim of the study is to use qPCR to identify potential biomarkers of HRSV and HBoV infection disease severity.

Major Comments:

  1. The authors do not provide information about the length of time between symptom onset, hospital admission and first sample collection and the interval between infection and sampling is unclear. Responses are presumably time-based- was there any evidence for this?

Reply: It would be very interesting information as suggested by the reviewer. Unfortunately, we don’t have information about the length of time between symptom onset, hospital admission, first sample collection, and the interval between infection and sampling.

  1. The secondary aim was to identify “whether the association of HRSV and HBoV would worsen the progression of the illness during HRSV and HBoV infections”, which was not addressed directly. The authors measured the levels of inflammatory mediators in patients with co-infection, but this was not associated with disease severity nor the progression of illness, particularly as the analysis is static and there is no mention of repeat sampling.

Reply; Reply; we would like to thank the reviewer for pointing out this observation, the associated between HRSV and HBoV co-infection has been added, as described on Table 4

Table 4. Comparison of cytokines and chemokines markers in the NPAs among children with HRSV and/or HBoV and disease severity

Children with HRSV

Children with HBoV

Children HRSV and HBoV co-infection

 Mild (n= 20)

Severe (n= 16)

 Mild (n= 18)

Severe (n= 5)

 Mild (n= 10)

Severe (n= 8)

INF-G

2.32(1.71-2.11)

3.46(0.39-4.32)

1.66(1.74-2.56)

1.46(1.38-2.18)

1.66(1.54-4.55)

2.42(0.19-4.20)

TNF-a

1.13(1.84-5.10)

2.51(0.47-6.80) *

1.60(0.44-4.50)

0.51(0.38-2.80)

3.60(1.46-9.50)

6.85(2.80-11.60) *

IL-1a

3.11(1.45-10.16)

3.91(2.44-34.86)

1.58(0.13-4.29)

1.87(1.44-33.9)

1.88(0.18-3.29)

2.92(2.26-5.42)

IL-2

15.5(4.39-31.0)

17.0(2.55-25.0)

18.0(2.84-33.50)

16.0(0.53-26.0)

3.0(2.84-4.50)

4.0(4.24-6.0)

IL-4

1.70(0.17-0.76)

1.48(0.33-0.56)

0.60(0.13-0.74)

0.50(0.33-0.54)

1.40(0.13-1.74)

1.32(0.13-1.66)

IL-6

1.66(0.40-4.32)

2.67(1.49-6.5) *

1.90(0.37-5.77)

1.67(0.49-2.43)

1.80(0.35-5.67)

0.71(0.50-1.87)

IL-8

1.60(0.5-3.07)

2.90(1.25-3.50) *

2.0(2.04-5.20)

1.80(0.27-1.16)

2.0(1.34-4.40)

2.27(0.20-2.9)

IL-10

0.60(0.20-2.67)

1.50(0.33-3.66) *

0.27(2.13-969)

2.69(15.26-29.87) *

7.22(0.18-0.66)

14.20(0.20-0.96) *

IL-13

1.13(0.70-15.3)

2.80(2.60-5.56) *

5.29(1.54-16.08)

3.90(0.50-1.56) *

5.27(1.54-17.08)

13.14(0.78-22.84) *

IL-17

0.16(0.18-0.75)

0.19(0.05-1.59)

1.37(0.13-0.76)

1.13(0.15-1.56)

1.37(0.13-1.76)

2.13(0.05-4.74)

IL-22

1.58(0.12-5.96)

3.15(1.55-3.60)

1.62(1.96-12.57)

2.19(1.34-13.57)

3.62(1.96-13.54)

282(1.13-14.40)

IL-33

9.55(3.30-10.33)

23.6(9.1-35.01) *

5.70(3.03-10.94)

20.6(7.1-33.01) *

5.73(3.06-11.94)

16.02(5.74-23.3) *

G-CSF

30.44(1.44-33.0)

29 (2.88-40.03)

3.27(1.28-33.41)

28.9(2.14-60.03)

1.27(1.28-3.41)

3.0(2.67-8.0)

CCL-2

10.72(1.66-21.45)

18.30(0.58-17.8)

6.95(0.10-9.90)

6.30(0.35-13.7)

3.95(0.14-5.90)

7.73(11.64-12.73)

CCL-3

1.28(1.71-4.127)

3.11 (0.35-14.11)

0.83(0.74-3.36)

2.33(0.35-23.02)

1.82(0.72-3.22)

2.04(2.28-14.62)

GM-CSF

0.18(1.73-1.71)

0.86(0.44-1.73)

1.74(0.54-2.81)

0.88(0.68-1.93)

1.74(0.52-2.82)

2.64(0.92-2.62)

CCL-4

1.11(0.23-13.4)

1.89(0.31-2.12)

1.10(0.11-28.6)

0.87(0.41-2.12)

2.13(0.13-8.0)

3.0(2.30-5.19)

CCL-5

1.11(0.07-1.44)

1.23(1.23-0.33)

1.10(0.07-1.82)

0.87(0.41-2.12)

1.10(0.07-2.82)

1.13(1.07-2.40)

Note: Data presented Median (25th -75th) percentile. * Denotes a significant difference between mild and HBoV of each virus respectively at 0.05 and 0.01 level.

  1. The tables include individual cases in several categories, presenting the same data repeatedly in different ways. This could be described as ‘reshuffling the deck’, looking for patterns each time the deck is redistributed. Statistical advice should be sought.

Reply; We agree with the assertation of the esteemed reviewer regarding this point, the data were represented as described on Table 1-4 with Statistical analysis. The non-parametric one-way analysis of variance (ANOVA) with posthoc Dennet’s test was used for data analysis.). Categorical data were presented as frequencies and percentages (%). Non-Gaussian variables were presented as the median. Independent Sample T-test and The Chi-Square independence test were used to indicate differences as all variables are nominal. The results with P < 0.05 were considered significant, page 4.                            

Additional points:

  1. How were these controls recruited and consented?

Reply; We don’t agree with the assertation of the esteemed reviewer. The controls recruited were presented as described on page 3.  Children who were reported positive for any other respiratory viruses were not included in the study. Control samples (n = 25) were collected from a group of healthy, asymptomatic age-matched children of whom NPA samples were negative for HRSV and HBoV using rRT-PCR.

  1. I could not see if all the datasets were from NPA or if some were serum/plasma. Please make this clear for each data table/graphic.

We don’t agree with the assertation of the esteemed reviewer. All of samples were nasopharyngeal aspirate samples (NPAs) as described on page 3. A total of 75 nasopharyngeal aspirate samples (NPAs) were previously identified using rRT-PCR assay by King Khalid University Hospital (KKUH) (36 positives HRSV, 23 positives HBoV and 16 HRSV coinfection with HBoV).

  1. Validation by direct measurement of selected protein level would provide confirmation.

Reply; We agree with the assertation of the esteemed reviewer. Moreover, we measure HRSV and/or HBoV at protein levels of proinflammatory cytokines and chemokines markers in future work to confirm these results, as mentioned on page 9. We acknowledge the shortcomings and limitations in terms of the methodology we used in our present study. Since it is easier to measure mRNA levels across the entire genome than protein levels, mRNA expression levels are frequently used as a proxy for estimating functional differences that occur at the protein levels [87]. Therefore, further work should be measured on HRSV and/or HBoV at protein levels of proinflammatory cytokines and chemokines markers for disease severity.

  1. The methods say that the subtyping of RSV-A and -B was done by measuring the RSV G gene by RT-qPCR, but the paper cited examined the RSV N gene.

Reply; We agree with the assertation of the esteemed reviewer, subtyping of RSV-A and -B was done by measuring the RSV G gene by one-step reverse transcriptase polymerase chain reaction (RT-PCR) assay as mentioned on page 3 and 4. The subgroup typing of HRSV-A and B was performed by amplifying the G gene using the same one-step RT-PCR kit along with subgroup-specific primers HRSV A-F (5′-GATGTTACGGTGGGGAGTCT-3′) and HRSV-A-R (5′-GTACACTGTAGTTAATCACA-3′) for group A viruses and HRSV B-F (5′-AATGCTAAGATGGGGAGTT-3′) and HRSV B-R (5′-GAAATTG AGTTAATGACAG-3′) for group B viruses [34].

  1. The numbers in the tables do not always add up as they should.

 Reply; We agree with the assertation of the esteemed reviewer, the tables have been corrected.

I recommend:

  1. Table 1 should be online.
  2. Table 2 can remain in the main text.
  3. Table 3-7 should be online, replaced in the main text by visualisations of distribution of significant mediator values. Showing these data as, for example, Violin plots would show individual datapoints and the spread of values between groups.

Reply; We agree with the assertation of the esteemed reviewer, the tables have been corrected. Table 1 was put in Supplementary data, the others tables were represented 

Reviewer 2 Report

In this work, Alkubaisi et al. evaluated the molecular profile of inflammatory mediators for two important controlled pediatric viral pathogens. The research aims to answer whether proinflammatory markers would have good predictive value for complications in the infection. From the results, some of these markers have potential.

 Analyzing the work, I believe that some issues deserve further deepening:

1)      Why not use real-time PCR for virus detection? There is a potential loss of sensitivity and the viral load (or even Ct value) would be an interesting data to be evaluated in comparison to inflammatory markers

2)      Study limitations should be included in the discussion

Author Response

Response to reviewers-2’ comments

We are very grateful for the reviews and valuable comments provided by the editors and each of the external reviewers of this MS

Reviewer 2

In this work, Alkubaisi et al. evaluated the molecular profile of inflammatory mediators for two important controlled pediatric viral pathogens. The research aims to answer whether proinflammatory markers would have good predictive value for complications in the infection. From the results, some of these markers have potential.

 Analyzing the work, I believe that some issues deserve further deepening:

  • Why not use real-time PCR for virus detection? There is a potential loss of sensitivity and the viral load (or even Ct value) would be an interesting data to be evaluated in comparison to inflammatory markers

Reply; We agree with the assertation of the esteemed reviewer. As a matter of fact, A total of 75 nasopharyngeal aspirate samples (NPAs) were previously identified using rRT-PCR assay by King Khalid University Hospital (KKUH) (36 positives HRSV, 23 positives HBoV and 16 HRSV coinfection with HBoV), as mentioned on page 3.

In addition, rRT-PCR, which is becoming the method used for respiratory diagnosis, yields result as cycle threshold (CT) values, which is a semi-quantitative value that can broadly categorize the concentration of viral genetic material in a patient sample, therefore, further research is required to examine the relationship between viral load or CT values and in-flammatory markers. as mentioned in page 9 and 10, and his comments will be valuable for our future studies.

2)      Study limitations should be included in the discussion

Reply; we would like to thank the reviewer for pointing out this observation, the limitations have been added, as described on page 9 and 10. We acknowledge the shortcomings and limitations in terms of the methodology we used in our present study. Since it is easier to measure mRNA levels across the entire genome than protein levels, mRNA expression levels are frequently used as a proxy for estimating functional differences that occur at the protein levels [87]. Therefore, further work should be measured on HRSV and/or HBoV at protein levels of proinflammatory cytokines and chemokines markers for disease severity. In addition, rRT-PCR, which is becoming the method used for respiratory diagnosis, yields result as cycle threshold (CT) values, which is a semi-quantitative value that can broadly categorize the concentration of viral genetic material in a patient sample, therefore, further research is required to examine the relation-ship between viral load or CT values and inflammatory markers. Since in vivo study is not sufficient for proving the induction of chemokines and cytokines by HRSV and/or HBoV, in vitro model is required to support the in vivo observations.

Reviewer 3 Report

This reviewer has two major concerns. First, the study is performed on clinical specimens, which in principle is a very good approach. However, it is not wise to make use of qPCR for the detection of cytokines and chemokines, as they cannot be thouroughly quantified by their respective mRNAs. The authors instead have to look for the proteins, as e.g. done in reference 69 (Khalfoui et al).

The limitation in the quantification in the present study is two-fold. On the one hand a high mRNA level does not necessarily mean that there is more protein, second the sampling of clinical specimens requires excellent and well controlled internal controls, or additional experimental appraoches in vitro which support the in vivo observations.

The second major concern was already mentioned with the last sentence, i.e. the authors have not tried to compare their data from the clinical cohort to in vitro data from cell cultures, which in turn would be crucial. As a minimal prerquistie they should demonstrate that they can quantify the chemokines and cytokines by qPCR, which requires a cell culture experiment controlled also by quantitative protein analyses, as e.g. done by Khalfaoui and coworkes.

Thus, despite a very good overall idea, the study fails to convincingly deliver novel insights, unless those controll experiments are made.

Author Response

Response to reviewers-3’ comments

We are very grateful for the reviews and valuable comments provided by the editors and each of the external reviewers of this MS

Reviewer 3

Comments and Suggestions for Authors

This reviewer has two major concerns. First, the study is performed on clinical specimens, which in principle is a very good approach. However, it is not wise to make use of qPCR for the detection of cytokines and chemokines, as they cannot be thouroughly quantified by their respective mRNAs. The authors instead have to look for the proteins, as e.g. done in reference 69 (Khalfoui et al).

Reply; We don’t agree with the assertation of the esteemed reviewer regarding this point. As a matter of fact, Typically, cytokines have been evaluated by serological-based approaches including enzyme-linked immunosorbent assay (ELISA), Luminex, and cytometric bead array (CBA) [28]. However, cytokine mRNA quantification is widely used to investigate cytokine profiles via molecular-based assays, particularly in small samples. In comparison with serological-based approaches, real-time reverse transcriptase polymerase chain reaction (rRT-PCR) is the method of choice for rapid and reproducible, and flexibility to measure cytokine from cells, tissues, or tissue biopsies [29-32], as mentioned in page 3, regarding this point, his comments will be valuable for our future studies.

The limitation in the quantification in the present study is two-fold. On the one hand a high mRNA level does not necessarily mean that there is more protein, second the sampling of clinical specimens requires excellent and well controlled internal controls, or additional experimental appraoches in vitro which support the in vivo observations.

As a matter of fact, since it is easier to measure mRNA levels across the entire genome than protein levels, mRNA expression levels are frequently used as a proxy for estimating functional differences that occur at the protein levels [87]. Therefore, further work should be measured on HRSV and/or HBoV at protein levels of proinflammatory cytokines and chemokines markers for disease severity, regarding this point, his comments will be valuable for our future studies. Page 9.

The second major concern was already mentioned with the last sentence, i.e. the authors have not tried to compare their data from the clinical cohort to in vitro data from cell cultures, which in turn would be crucial. As a minimal prerquistie they should demonstrate that they can quantify the chemokines and cytokines by qPCR, which requires a cell culture experiment controlled also by quantitative protein analyses, as e.g. done by Khalfaoui and coworkes.

We thank the reviewer for his good comments and his comments will be valuable for our future studies. Since in vivo study is not sufficient for proving the induction of chemokines and cytokines by HRSV and/or HBoV, in vitro model is required to support the in vivo observations, page 10

Thus, despite a very good overall idea, the study fails to convincingly deliver novel insights, unless those control experiments are made.

We agree with the reviewer comment. Although the quantitative protein of the chemokines and cytokines and controlled internal controls through in vitro using cell lines are needed, we provide evidence that some cytokines and chemokines are a potential biomarker to monitor the severity of HRSV and HBoV infections in children using qPCR-based detection.

Round 2

Reviewer 1 Report

1. We think the wording should be changed to state that the aim of the paper is not to identify potential biomarkers to predict disease progression (which the Authors then say they do in the conclusions). Instead, this paper looks at inflammatory genes; they show the response to HRSV or HBoV, or co-infection, compared to healthy controls; this could provide some mechanistic insight into the difference in immune response between the two viruses but do not identify biomarkers for disease progression.

2. Under ‘Author Contributions’ it is stated that the corresponding author (Prof. Almajhdi) was responsible for funding acquisition. The manuscript should state the funding/grant under the Funding tab.

3. The authors have chosen not to respond to our suggestion that graphical representations of the data should replace some of the tables. The tables are not easy to digest as they contain a lot of information. A set of violin plot would better show the median and range of data for each significant mediator.

4. The authors should clarify the ethics and site under which their healthy controls were recruited.

Minor edits needed

Author Response

Response to reviewers’ comments

We are very grateful for the reviews and valuable comments provided by the Reviewer of this MS.

Reviewer 1

Comments and Suggestions for Authors

  1. We think the wording should be changed to state that the aim of the paper is not to identify potential biomarkers to predict disease progression (which the Authors then say they do in the conclusions). Instead, this paper looks at inflammatory genes; they show the response to HRSV or HBoV, or co-infection, compared to healthy controls; this could provide some mechanistic insight into the difference in immune response between the two viruses but do not identify biomarkers for disease progression.

Reply; We agree with the assertation of the esteemed reviewer, the objective of the study among MS has been corrected, page 3

Therefore, the primary purpose of the current study is to identify the change in inflammatory mediators during HRSV, HBoV, and HRSV coinfection with HBoV.

  1. Under ‘Author Contributions’ it is stated that the corresponding author (Prof. Almajhdi) was responsible for funding acquisition. The manuscript should state the funding/grant under the Funding tab.

Reply; we would like to thank the reviewer for pointing out this observation, the funding/grant has been added, as described on page 10.

Funding: The authors thank the Researchers Supporting Project number (RSPD2023R801), King Saud University, Riyadh, Saudi Arabia.

  1. The authors have chosen not to respond to our suggestion that graphical representations of the data should replace some of the tables. The tables are not easy to digest as they contain a lot of information. A set of violin plot would better show the median and range of data for each significant mediator.

Reply; we would like to thank the reviewer for pointing out this observation, Table 4 has been replaced with Figure 1, as described on page 7.

 Figure 1. Comparison of cytokines and chemokines markers in the NPAs among children with HRSV and/or HBoV and disease severity. Note: Data presented Median (25th -75th) percentile. * Denotes a significant difference between mild and HBoV of each virus respectively at 0.05 level.

  1. The authors should clarify the ethics and site under which their healthy controls were recruited.

Reply; we would like to thank the reviewer for pointing out this observation, the healthy controls have been clarified as described on page 3.

The healthy control NPAs samples (n = 25) were collected from infants without ALRTI symptoms and age-matched children of whom NPA samples were negative for HRSV and HBoV using rRT-PCR.

Reviewer 3 Report

The authors have thouroghly addressed my comments and transparently discussed the limitations of there study. The study as it stands now can be a good starting point for further research.

Author Response

We thank the reviewer for his recommendations and there are no other additions